# Heterogeneous Impacts of Body Mass Index on Work Hours

**DOI:** 10.3390/ijerph18189849

**Published:** 2021-09-18

**Authors:** Young-Joo Kim

**Affiliations:** Department of Economics, Hongik University, Seoul 04066, Korea; y.j.kim@hongik.ac.kr

**Keywords:** body mass index, obesity, work hours, threshold

## Abstract

This study examined how higher body mass index (BMI) affects the work hours of men and women and how the impact varies by gender and the value of BMI. Using a longitudinal dataset of 1603 British adults (men: *n* = 775; women: *n* = 828) and a panel threshold regression model, this study estimated that BMI has significant impacts on work hours but the pattern is different by gender and BMI groups. BMI is positively associated with work hours up to the estimated BMI threshold of 30, which corresponds to the clinical cutoff point of obesity; above this point, additional increases in BMI is associated with reduced work hours. The asymmetric nonlinear relationship between BMI and work hours was more evident among women, particularly female low-skilled workers. The results imply reduced work capacity and lower labor income for women with a higher BMI above an obesity threshold, highlighting a practical role of BMI’s obesity cutoff value. The findings of this study provide a new perspective regarding the economic burden of workplace obesity and point out the need to design gender-specific and BMI-based strategies to tackle productivity loss from obesity.

## 1. Introduction

With a rapid increase in obesity worldwide, we are seeing increasing scholarly attention paid to its detrimental economic impact. The adult obesity rate increased from 15.6 percent to 39.8 percent between 1988 and 2016 in the United States and from 15 percent to 26 percent between 1993 and 2016 in England [1,2,3]. This increased obesity prevalence has reduced labor market participation and the disability-free life years of workers, consequently decreasing the productive workforce and the potential economic output [4,5]. The average productivity loss arising from obesity is projected to lower gross domestic product by 3.3 percent in developed countries including the US and UK between 2020 and 2050 [4]. It is worrisome that obesity’s total economic cost exceeds medical costs [5] and indirect costs arise from consequent productivity losses. Owing to the adverse health conditions associated with obesity and being overweight [6,7,8,9], various forms of productivity loss, from a reduction in wages to health-related limitations at work (presenteeism), absence from work (absenteeism), disability and premature mortality, can occur. According to Dee et al. [10], such indirect costs account for 60 percent of the total costs of being overweight or obese. Indirect costs are more difficult to measure than direct costs and yet, they comprise a crucial component of obesity costs. Therefore, it is important to obtain a systematic assessment of the indirect costs of obesity in this era of increasing obesity prevalence.

Numerous studies have examined the indirect costs of obesity. Among the potential components of such costs, absenteeism and presenteeism have been estimated to contribute the most to lost workplace productivity [11,12]. Specifically, high body mass index (BMI)s are associated with reduced work capacity and increased absenteeism and presenteeism [13,14]. Other studies have examined the impact of obesity on wages for assessing the indirect cost of obesity. Using estimation methods based on linear regression models, Baum and Ford [15], Cawley [16] and Morris [17] found negative effects of BMI on wages, particularly for women in the UK and the US. Later studies that adopted more general functional forms of estimation methods also found a negative relationship between BMI and wages, but in a nonlinear manner such that workers within a certain BMI range received lower wages than those within healthy ranges. See, for example, the studies by Kline and Tobias [18], Gregory and Ruhm [19] and Kan and Lee [20]. Therefore, a consensus has been reached, at least for women, regarding the negative relationship between BMI and wages.

Work hours are another factor representing workplace productivity. Adverse health conditions associated with excess body weight may affect work hours during normal working days. Furthermore, the overall impact of obesity on labor income depends on both the hourly wage rate and hours of work; however, evidence for the relationship between obesity and work hours is scarce. In particular, the question of how obesity is associated with average work hours beyond sick days remains unaddressed. There are several advantages to the use of work hours as an alternative measure of workplace capacity. First, work hours are reported without any subjective evaluation of one’s own work performance, which improves the accuracy of its measurement. Second, work hours are easy to compare to assess variations in work performance during an individual’s work life. Third, work hours have already been collected in some longitudinal studies, which enables us to control for individual-specific fixed effects.

This study examined how productivity loss from obesity differed by gender using work hours as a primary productivity measure. Some studies reported gender differences in size and pattern of productivity loss from obesity. For instance, Finkelstein et al. [11] showed a higher cost of obesity in terms of absenteeism and presenteeism for women than men. Cawley [16] found significant costs of obesity in wages only among women whereas Kline and Tobias [18] reported a greater loss in wages for men than women from high BMI and obese groups. Evidence for different associations between BMI and work productivity by gender implies the need for gender-specific approaches in quantifying the obesity cost of any productivity measure.

In evaluating obesity cost, we need to note that obesity is observed only in the upper part of the BMI distribution and thus the relationship between BMI and workplace productivity can differ along the BMI distribution. Findings from previous literature, including Kan and Lee [20] and Kline and Tobias [18], support this notion by showing the nonlinear aspect of productivity loss along BMI distribution. Gates et al. [14] noted the threshold effect of BMI on presenteeism, such that BMI over a certain threshold has a greater negative impact than a lower BMI below the threshold on presenteeism. To capture such potential heterogeneity in the magnitude of productivity loss over different BMI value, this study adopted a novel approach of generalized method of moments (GMM) estimator of the panel threshold model which has been newly developed in the literature. This method provides the estimates of both a threshold point and the details of how BMI and work hours are related below and above this threshold in the presence of a nonlinear relationship. The specific approach for detecting a BMI threshold, at which the BMI’s effect on work hours changes, allows us to distinguish the most vulnerable group and set an attainable BMI target to achieve the highest benefit in health and economic costs from this reduction in BMI. Therefore, the next task is to define the potential threshold effect of BMI on work hours, but we lack such evidence.

It is important to examine multifaceted factors to comprehensively explain the economic cost of obesity; furthermore, it would be beneficial to identify a threshold effect of BMI alongside gender-specific effects. By using longitudinal data and a panel-based estimation method, we derive a better understanding of obesity’s effect on work capacity, which is a crucial component of indirect costs. Accordingly, the purpose of this study is to examine the impact of obesity on the work hours of men and women over their careers using an empirical framework that can assess and reveal a nonlinear relationship, if any, between BMI and work hours.

## 2. Materials and Methods

### 2.1. Data Collection

The data for this study were taken from the 1970 British Cohort Study (BCS), which surveyed all the people who were born in England, Scotland and Wales in one particular week in 1970 (see Brown [21] and Elliott and Shepherd [22] for details). For this study, information on health and primary labor market outcomes was taken from the 2004 and 2012 surveys at a time when the respondents had reached 34 and 42 years of age. For background details, the individuals’ data collected at birth and at age 10 years were used. The BCS data were collected through face-to-face interview and self-completion.

Among the 2819 participants who had completed all questions related to health and labor market outcomes, this study focused on 1753 participants who worked full- or part-time during both 2004 and 2012 and were not self-employed as of 2004. In the female sample, 68 participants were excluded as they had been pregnant in one of the survey years. In the BCS, a “work hour” for workers who were not self-employed was defined as the weekly work hours, including normal hours and overtime, for their main job. We focused on participants who worked at least 10 h and less than or equal to 70 h, which was the 99th percentile of the work hour distribution, excluding 82 workers. The final sample over two years are 1656 observations from 828 female workers and 1550 observations from 775 male workers. The sample selection process is illustrated in Figure 1.

Participants’ BMIs in adulthood were derived by the BCS based on self-reported weights and heights from the interviews. Various measures of individual characteristics, including education level, English residence and family size, were used in the model. Occupation categories were divided into low-skill and high-skill jobs based on the BCS social class grouping. A high-skill occupation was defined as a professional, technical, or managerial job. Some of the jobs listed in this occupation category are chief executive, doctor, lawyer, nurse, actor, journalist, engineer and teacher.

Childhood and parental weight variables were used as instruments for adult BMI for deriving causal effect of BMI on work hours. The weight measures are the respondent’s birth weight and the BMI values of the individual and his or her mother and father when the respondent was ten years old.

### 2.2. Statistical Model

We employ the panel threshold regression model; a worker in the high BMI group may respond to a BMI increase with more work hour sensitivity than one in the normal group. Since the threshold that classifies high BMI group from normal group is unknown a priori and whether this threshold remains the same across different skill level and gender groups, it is estimated from the data. In other words, the model below allows BMI’s impact on work hours to change at an unknown threshold point while addressing unobserved individual heterogeneity.
(1)$$y_{it}=\beta_{0}+x_{1it}\beta_{1}+\cdots +x_{kit}\beta_{k}+q_{it}\alpha +\delta \left(q_{it}-\gamma \right)1\left\{q_{it}\geq \gamma \right\}+\omega_{i}+\varepsilon_{it}.\;$$
where $y_{it}$ is the weekly work hours for an individual i at age 34 (t = 1) or age 42 (t = 2) years, $q_{it}$ is the BMI for each period t and $x_{1it}$ through $x_{kit}$ are other time-varying individual characteristics. $\omega_{i}$ is an individual fixed effect and γ is the threshold parameter at which the slope of BMI changes according to the nonlinear relationship between BMI and work hours. Here, δ denotes the change in the slope of BMI at an unknown BMI point γ and the specification of $\delta \left(q_{it}-\gamma \right)1\left\{q_{it}\geq \gamma \right\}$ ensures that the change occurs continuously without a jump.

To control for potential endogeneity of the threshold variable BMI, which results from its correlation with $\omega_{i}+\varepsilon_{it}$, Equation (1) is transformed through first-differencing to obtain the following model of Equation (2):(2)$$Ɗy_{i2}=Ɗx_{1i2}\beta_{1}+\cdots +Ɗx_{ki2}\beta_{k}+Ɗq_{i2}\alpha \;+\delta \left[\left(q_{i2}-\gamma \right)1\left\{q_{i2}\geq \gamma \right\}-\left(q_{i1}-\gamma \right)1\left\{q_{i1}\geq \gamma \right\}\right]+Ɗ\varepsilon_{i2}$$
where $Ɗy_{i2}=y_{i2}-y_{i1}$, $Ɗx_{ji2}=x_{ji2}-x_{ji1}$ and $Ɗ\varepsilon_{i2}=\varepsilon_{i2}-\varepsilon_{i1}$.

We are interested in estimating the BMI impact α and α+δ, which represent the slopes in the regimes of $q_{it}<\gamma$ and $q_{it}\geq \gamma$, respectively and an unknown threshold γ.

### 2.3. Estimation Methods

As a benchmark analysis, BMI and work hours’ linear relationship was first examined with the restriction that δ=0. We used two approaches to estimate linear models: pooled ordinary least squares (OLS) and fixed effects estimation methods.

Next, we investigated a possible nonlinear relationship by easing the restriction δ=0. Recently Seo and Shin [23] developed the GMM estimator of the dynamic threshold model based on first-difference transformation for estimating the parameters in a nonlinear panel threshold model. The current study used the GMM estimation procedure developed in Seo and Shin [23] and Seo et al. [24]. The instruments for the BMI ($q_{it})$ included birth weight, BMI at age 10 and the BMIs of the respondent’s mother and father when the respondent was ten years old. A weak instrument test [25] and an overidentification test were performed in support of these instruments.

The unknown threshold γ was drawn from the interior of the support of the BMI distribution; that is, some fraction of the smallest and largest values of the sample BMI observations was trimmed in estimating the threshold. This was done to reduce the noise caused when a threshold is picked up from extreme values at the tails of the distribution; however, too much trimming can exclude the true threshold point. We trimmed 15 percent of the sample from each side of the BMI distribution. For sensitivity analysis, the threshold estimates from different trim rates were examined.

## 3. Results

Table 1 depicts the summary statistics for 2004 and 2012 when the respondents reached 34 and 42 years of age, respectively. There is a clear difference in working hours between men and women. The average hours worked per week were 45 for men and between 31 and 33 for women. Men tend to work more than 12 h compared to women at age 34 and 42 but the variation in work hours is larger in women.

The average BMI was approximately 25 for women and 27 for men, with a slightly larger variation in BMI among women than men in both years. For controls of individual characteristics that are closely related to work hours, five variables were included: having at least a bachelor’s degree, residing in England, having a high-skill occupation, the number of household members and age. As shown in Table 1, education level, family size and occupation level increased for both men and women from 2004 to 2012. Regarding the gender differences in these covariates, education level is similar, but the proportion of high-skilled workers is higher in men than women.

In Table 1, the lower panel depicts the childhood weight variables that were used as instruments for BMI in adulthood. That is, the respondent’s birth weight and the BMIs of the individual and his or her mother and father when the respondent was 10 years old. These childhood weight variables were strongly correlated with and explained 20 percent of BMI variation in adulthood. However, these instruments were not associated with work hours once BMI at either 34 or 42 years of age and other individual characteristics were controlled for with the *t* statistic being between −0.98 and 1.37. The weak instrument test (with the Cragg–Donald Wald *F* statistic at 167.83 and the Stock–Yogo critical value at 16.85) and overidentification test (with the *p*-value of the Sargan statistic at 0.649) also supported the instruments’ validity. These are in line with Kline and Tobias’s [18] findings from the analysis of the BCS that parental BMIs are good predictors of an individual’s BMI in adulthood.

### 3.1. Relationship between Work Hours and BMI

Table 2 presents the estimated BMI effect on work hours from the three different methods. The first two columns are from linear models of pooled OLS and fixed effects regressions. The upper panel is for female workers and the lower panel for male workers. For female workers in column 1, we can see that BMI is positively associated with work hours; a higher BMI is associated with longer work hours. However, once we control for unobserved individual characteristics that are constant over time using fixed effects regression, there appeared to be no significant relationship between BMI and work hours.

The results in column 3 from the GMM estimator of the panel threshold model depict a different picture of the BMI effect on work hours. The marginal effect of BMI changes at the estimated threshold BMI value; a higher BMI is associated with longer work hours up to the threshold of 30.02, above which a further increase in BMI is associated with decreased work hours. Specifically, estimates of the BMI effect below the threshold (α) and the additional BMI effect above the threshold (δ) were 4.66 and −7.09, respectively; they were both statistically significant. That is, in response to a marginal increase in BMI below the threshold of 30, work hours increased by 4.66 and thereafter work hours decreased by 2.43. Another notable point is that the estimated threshold point γ was 30.02, which is close to the clinical cutoff point of obesity. The results suggest that there exists a heterogeneous BMI effect on work hours along the distribution of BMI. Based on the results, we can infer that the nonlinear relationship between BMI and work hours averaged out to be positive or zero when the linear regression models were used.

In the lower panel of Table 2 for male workers, we can see that BMI is not significantly associated with men’s work hours in either pooled OLS or fixed effects estimation methods. The GMM panel threshold results in column 3 also showed that BMI is not significantly associated with men’s work hours although the BMI threshold point was estimated to be 25.66.

#### 3.1.1. Sensitivity Test

The presence of the threshold effect for female workers is a critical feature of the current findings. Further analyses were conducted for sensitivity tests in the following two subsections. First, different trim rates were used in the threshold point estimation. Second, the GMM estimator of the panel threshold model was applied to the sample after workers were divided into high-skilled and low-skilled workers. For a graphical illustration of the key features of work hours and BMI, Figure 2 and Figure 3 present the kernel density of each of these variables by skill level. Figure 2 is for female workers and Figure 3 for male workers.

#### 3.1.2. Female Workers: BMI and Work Hours

The upper panel of Table 3 presents the threshold effect of BMI on work hours and the threshold point with different trim rates. When the trim rate was increased from 15 to 20 percent from each side and, thereby, more observations from the tails of the distribution were excluded in the grid search of a threshold point, the estimated threshold decreased from 30.021 to 28.277 and was statistically significant. For a lower trim rate of 10%, the threshold point was 30.955. Thus, with the trim rates shifting from 10 to 15 and 20% from each side of the distribution, the threshold point varied between 28.277 and 30.955, tightly containing the obesity cutoff of 30.

The lower panel of Table 3 presents the estimated threshold effect of BMI by worker’s skill level. Two findings were obtained from this analysis. First, the threshold point was estimated to be approximately 30 for both skill levels, 30.02 for high-skilled workers and 30.54 for low-skilled workers. Second, the pattern of nonlinear effects differed by skill level. For high-skilled workers, a significant effect of BMI emerged only for BMIs above the threshold. Specifically, there was no significant relationship between work hours and BMI up to the threshold of 30.02, whereas a further increase in BMI above the threshold had a significant negative effect. For low-skilled workers, significant and contrasting patterns were observed both below and above the threshold. A marginal increase in BMI up to the threshold of 30.54 had a positive effect on work hours, whereas a marginal increase in BMI beyond the threshold had a negative effect on work hours.

The threshold BMI effect contingent on worker’s skill levels can be briefly assessed through comparison of the distributions of BMI and work hours by skill level. Figure 2 shows that, among women, there is a clear difference in the distribution of work hours between low- and high-skilled workers. Full-time work with at least 35 h per week is more common among high-skilled than low-skilled workers. Specifically, approximately 63 percent of high-skilled female workers perform 35 or more hours of work. Conversely, only 36 percent of low-skilled workers perform 35 or more hours. Figure 2 also illustrates that the BMI distribution is more dispersed among low-skilled than high-skilled workers and that the healthy BMI range includes a larger fraction of high-skilled workers than low-skilled ones.

#### 3.1.3. Male Workers: BMI and Work Hours

The threshold parameter for male workers was estimated to be approximately 25.6, which is close to the overweight cutoff and is well below the threshold of 30 for female workers, as shown in the lower panel of Table 2. For the first sensitivity check, the threshold BMI effect was examined with different trim rates in Table 4. The threshold point of BMI is robust around 25.6 across the trim rates of 10 to 20 percent from each tail of the distribution. Despite a consistent and statistically significant estimate of the threshold point, there was no clear impact of BMI on the hours worked by male workers either below or above the BMI threshold. Regardless of the BMI value, male workers consistently worked long hours in their thirties and forties, as indicated by the stable working hour pattern expressed in Table 1. For the next sensitivity test, the BMI effect was examined by skill level. As presented in the lower panel of Table 4, there was no significant effect of BMI on the hours worked by male workers for either skill level although the threshold points were estimated to be significant for both skill levels with a higher threshold for low-skilled workers. Figure 3 depicts the distributions of male workers’ work hours and BMI by skill level. Consistent with the results in Table 4, the work hour distribution is very similar between the two skill groups, except that a larger fraction of high-skilled workers supplied longer hours than low-skilled workers. In addition, the BMI distribution is similar across skill levels although high-skilled male workers are more likely to be in the healthy BMI range than low-skilled workers, as among female workers.

## 4. Discussion

This study examined the impact of and work hour responses to changes in BMI for men and women during their careers at ages 34 and 42 years. This study provided new evidence of the nonlinear relationship between BMI and work hours by demonstrating a BMI threshold effect. BMI was positively associated with work hours up to the threshold BMI value of 30, beyond which a negative BMI effect dominated. The threshold effect is more evident for women, particularly those with low-skilled jobs that were more likely to require physical strength. The total effect of BMI on work hours above the threshold was −2.43. Since the standard deviation of average work hours was 11.68, the BMI effect on work hours above the threshold amounted to a decrease in work hours by 21 percent of a standard deviation. This negative effect of BMI for BMI above the threshold was robust across female workers with different skill levels.

The notable finding is that the nonlinear impact of BMI is evident in women but not in men. Two possible explanations are proposed here. One reason for the insignificant effect of BMI on men’s work hours is the relatively small variation in work hours among men. Most men worked at least 35 h regardless of skill level and only a few (2.68%) worked less than 35 h. Half of women (50.36%), on the other hand, worked less than 35 h. Therefore, in the presence of restricted work hour pattern for male workers, it is plausible that men are less likely to respond to an increase of BMI and a consequent change in health status compared to women. The other possible explanation for different BMI impact by gender is biological differences between men and women. Women may respond differently than men to changes in health conditions associated with obesity. For example, the majority of hospital episodes for bariatric surgery in England of the UK has been taken up by women between 2000 and 2012 (e.g., 82.4% in 2000–2001 and 76.3% in 2011–2012) according to Gatineau et al. [6].

The results of nonlinear relationship between BMI and work hours and differences in the BMI impact across gender are in line with previous findings of the BMI impact on other productivity measures. For example, studies by Gregory and Ruhm [19], Kan and Lee [20] and Pinkston [26] examined workers in the US and showed negative effects of BMI on hourly wages for women in particular with evidence of nonlinearity. Kline and Tobias [18] also found negative nonlinear impact of BMI on wages of men and women in the UK. Several studies around the globe also found significant relationships between BMI in the obesity range and work-related limitations including absenteeism and presenteeism. DiBonaventura et al. [27], Finkelstein et al. [11] and Gates et al. [14] examined workers in the US, Bustillos et al. [13] for Canada and Kudel et al. [28] for Brazil to document the negative impact of BMI or obesity on workplace productivity.

This study’s findings have several implications. First, given the previous evidence of negative BMI impact on hourly wages, the adverse effects of higher BMI on work hours suggest that a higher BMI can doubly increase the economic burden, as hourly wages and work hours determine labor income. A recent study from Denmark corroborates this notion. Kjellberg et al. [29] found that a higher BMI above 30 was negatively associated with mean annual income. Second, the positive effect of BMI on work hours below the threshold for low-skilled female workers may reflect the beneficial effects of muscle mass; Wada and Tekin [30] noted positive muscle mass impact on wages. Third, the gender and skill level-specific threshold of BMI broadly corresponded with the clinical obesity cutoff suggested by the World Health Organization [31], substantiating BMI’s important role in classifying excessive body weight across different groups. In this sense, the current study provides practical guidelines based on BMI thresholds and job skill level to identify workers who are at greater risk of bearing economic burden from obesity. Implementing public health policies that target the most vulnerable group of workers through proper management of gender-related health problems will effectively minimize obesity costs in the workplace. Finally, this study focused on work hours based on British data, but the threshold approach to detect nonlinear relationship can be applied to various health or economic outcomes in other part of the world including Asia, where obesity is an emerging health issue.

Despite this, study’s contributions described above, it has some limitations. First, the BCS data of height and weight were collected through interview and thus the data may contain measurement error although the reported height measures were consistent across the surveys. Second, workers were divided based on their occupation category rather than job characteristics such as physically demanding job or sedentary office job. The limited information restricts further analysis of disclosing the mechanism behind the BMI impact on work hours. Third, the sample was relatively small and it was restricted to workers who actively participated in the labor market, but one can pursue other measures such as employment years following Bokerman et al. [32]. Fourth, comparing voluntary and non-voluntary work hours can be useful with available data for understanding the underlying reasons for the changing BMI effects. Lastly, having a detailed history of respondents’ health conditions would enable us to identify the health risk factors associated with BMI above the threshold.

## 5. Conclusions

This study extends the literature on obesity’s cost by showing that BMI is differently associated with work hours depending on the level of BMI and that the pattern of nonlinear relationship is more evident among female workers. Specifically, BMI is positively associated with work hours only up to a threshold, particularly for low-skilled workers, above which a further BMI increase reduced work hours of women. These findings provide a new perspective on the economic workplace burden of widespread obesity and the need to develop active coping strategies to alleviate the overall costs of obesity.

## Figures and Tables

**Figure 1 ijerph-18-09849-f001:**
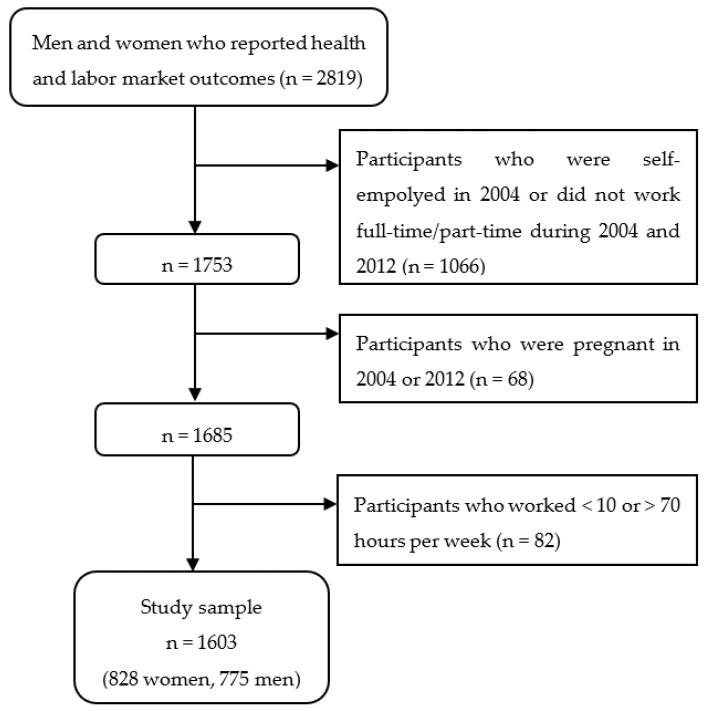
Flow diagram of the study sample selection.

**Figure 2 ijerph-18-09849-f002:**
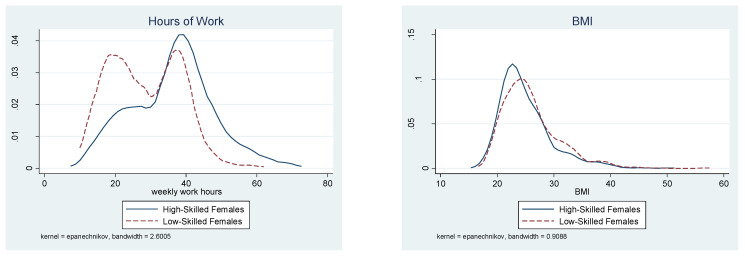
Distributions of Work Hours and BMI for Women by Skill Level.

**Figure 3 ijerph-18-09849-f003:**
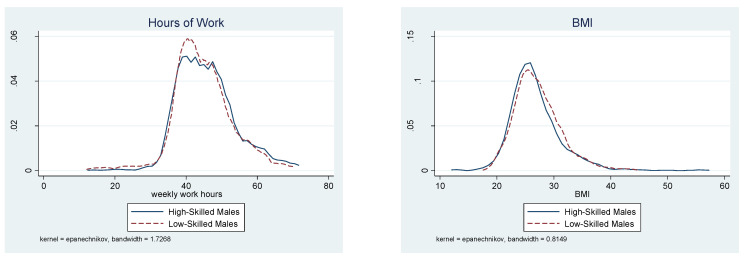
Distributions of Work Hours and BMI for Men by Skill Level.

**Table 1 ijerph-18-09849-t001:** Summary Statistics.

Variables	Mean (Standard Deviation)			
	Women (*n* = 828)		Men (*n* = 775)	
Adulthood Variables	Age 34	Age 42	Ag34	Age 42
Hours worked per week	31.343 (11.684)	33.534 (11.397)	45.285 (7.896)	45.561 (8.452)
BMI	24.919 (4.615)	25.891 (4.870)	26.579 (4.140)	27.334 (4.359)
Bachelor’s degree+ ^§1^	0.197 (0.398)	0.287 (0.453)	0.222 (0.416)	0.308 (0.462)
Family size	3.388 (1.006)	3.496 (1.026)	3.151 (1.039)	3.485 (1.140)
High-skill job ^§2^	0.469 (0.499)	0.551 (0.498)	0.533 (0.499)	0.578 (0.494)
Resident of England ^§3^	0.843 (0.364)	0.842 (0.365)	0.834 (0.373)	0.832 (0.374)
Childhood Variables				
Birthweight in kg	3.267 (0.506)		3.401 (0.521)	
BMI at 10	16.848 (2.160)		16.738 (1.872)	
Mother’s BMI at 10	23.248 (3.629)		23.338 (3.784)	
Father’s BMI at 10	24.489 (2.924)		24.340 (2.710)	

Notes: Sample size is 828 women and 775 men. ^§^ Dummy variables: Reference groups: ^1^ = have a bachelor’s or higher degree; ^2^ = have a high-skill job (professional, technical, or managerial job); ^3^ = reside in England. BMI is body mass index (kg/m^2^).

**Table 2 ijerph-18-09849-t002:** BMI Effect on Work Hours by Gender from Linear and Nonlinear Models.

**Panel A**	**Estimated BMI Effect on Work Hours from Three Models: Women**		
	**Pooled OLS**	**Fixed Effects**	**Panel Threshold**
	**(1)**	**(2)**	**(3)**
BMI effect (α)	0.254 ** (0.052)	0.026 (0.141)	4.656 ** (2.086)
Additional BMI effect over threshold (δ)			−7.088 * (3.854)
BMI threshold (γ)			30.021 *** (2.920)
*R^2^*	0.270	0.214	
Sample size	1656	1656	1656
**Panel B**	**Estimated BMI Effect on Work Hours from Three Models: Men**		
	**Pooled OLS**	**Fixed Effects**	**Panel Threshold**
	**(1)**	**(2)**	**(3)**
BMI effect (α)	0.025 (0.049)	0.137 (0.136)	−3.021 (3.643)
Additional BMI effect over threshold (δ)			3.439 (3.440)
BMI threshold (γ)			25.658 *** (2.250)
*R^2^*	0.01	0.01	
Sample size	1550	1550	1550

Notes: Sample size for women is 1656 observations from 828 individuals and for men it is 1550 from 775 individuals. Pooled OLS is ordinary least squares estimates from panel data. All models include controls of age, family size, education, occupation and region of residence. Standard errors are in parentheses. *** *p* < 0.001, ** *p* < 0.05, * *p* < 0.1.

**Table 3 ijerph-18-09849-t003:** BMI Effect on Work Hours using Panel Threshold Regression: Women.

**Panel A**	**Threshold BMI Effect on Work Hours with Different Trim Rates**		
	**All**	**All**	**All**
Key Variables	(1)	(2)	(3)
BMI effect (α)	4.650 ** (1.994)	4.656 ** (2.086)	5.566 ** (2.345)
Additional BMI effect over threshold (δ)	−6.730 ** (3.110)	−7.088 * (3.854)	−6.065 (4.335)
BMI threshold (γ)	30.955 *** (2.317)	30.021 *** (2.920)	28.277 *** (2.383)
trim rate	0.2	0.3	0.4
Sample size	1656	1656	1656
**Panel B**	**Threshold BMI Effect on Work Hours by Worker’s Skill Level**		
	**All**	**High-Skill Job**	**Low-Skill Job**
	**(1)**	**(2)**	**(3)**
BMI effect (α)	4.656 ** (2.086)	2.620 (2.746)	9.420 * (5.667)
Additional BMI effect over threshold (δ)	−7.088 * (3.854)	−7.088 * (3.854)	−12.578 * (7.096)
BMI threshold (γ)	30.021 *** (2.920)	30.021 *** (2.920)	30.540 *** (1.486)
Sample size	1656	1018	638

Notes: Sample size is 1656 observations from 828 individuals. All models include controls of age, family size, education, occupation and region of residence. Standard errors are in parentheses. *** *p* < 0.001, ** *p* < 0.05, * *p* < 0.1.

**Table 4 ijerph-18-09849-t004:** BMI Effect on Work Hours using Panel Threshold Regression: Men.

**Panel A**	**Threshold BMI Effect on Work Hours with Different Trim Rates**		
	**All**	**All**	**All**
Key Variables	(1)	(2)	(3)
BMI effect (α)	−3.056 (3.937)	−3.021 (3.643)	−3.042 (3.645)
Additional BMI effect over threshold (δ)	3.468 (3.606)	3.439 (3.440)	3.457 (3.442)
BMI threshold (γ)	25.630 *** (2.525)	25.658 *** (2.250)	25.642 *** (2.229)
trim rate	0.2	0.3	0.4
Sample size	1550	1550	1550
**Panel B**	**Threshold BMI Effect on Work Hours by Worker’s Skill Level**		
	**All**	**High-Skill Job**	**Low-Skill Job**
	**(1)**	**(2)**	**(3)**
BMI effect (α)	−3.021 (3.643)	0.299 (8.680)	−0.946 (2.056)
Additional BMI effect over threshold (δ)	3.439 (3.440)	−0.665 (6.165)	4.976 (5.774)
BMI threshold (γ)	25.658 *** (2.250)	25.036 (34.34)	31.242 *** (3.667)
Sample size	1550	996	554

Notes: Sample size is 1550 observations from 775 individuals. All models include controls of age, family size, education, occupation and region of residence. Standard errors are in parentheses. *** *p* < 0.001.

## Data Availability

The 1970 British Cohort Study is available from the UK Data Service. Stata code is available from the author’s webpage.

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
