# Peer review of "Heterogeneous Impacts of Body Mass Index on Work Hours"

_ijerph, 2021, doi:10.3390/ijerph18189849_

Round 1

Reviewer 1 Report

The Authors decided to explore the association between BMI and work time. It is an interesting approach to the socioeconomic burden of overweight and obesity, which is not often taken in literature. The study population included a relatively small, but acceptable number of individuals and was well balanced between the sexes.

Detailed comments regarding particular parts of the text are provided below:

  • Abstract:
    • line 17 – “and point upthe need to” – maybe it should be ‘’point out”;
  • Introduction:
    • 64-65 – there is a statement suggesting that the work hours were used as an “additional productivity measure”; however, the title of the article suggests that this was the main measure analysed in the study; thus, in the Reviewer’s opinion, writing it in a more precise way would be helpful;
  • Material and methods:
    • in the Reviewer’s opinion the term “high skilled jobs” should be defined more precisely; maybe some examples of jobs included in this category will help the reader to fully understand the categorisation into this group;
    • data on body height and weight of the participants were reported by the participants themselves, which is a significant limitation of the study; it should be discussed in the “Discussion” part of the text; additionally, there should also be information in “Material and methods” part of the text, regarding how exactly the information on body height was collected (i.e. over the phone/ internet survey, in-person etc.);
    • another limitation of the study is the fact, that the participants were not divided based on the character of their jobs – more vs less physically demanding; there will be a significant difference between physically demanding jobs (e.g. building, carrying heavy items) and activities associated with office-type jobs in the context of the study;it should be discussed in the “Discussion” part of the text;
    • in the Reviewer’s opinion, the process of inclusion/ loss of participants from the study group will be clearer for the reader if it is presented in the form of a diagram rather than a full description in the text.

Author Response

Point-by-point responses to Reviewer 1

I thank the referee for reviewing of my paper "Heterogeneous Impacts of Body Mass Index on Work Hours." I appreciate the helpful feedback from the reviewer. I carefully addressed the reviewer’s comments and suggestions and incorporated them in the revised version of my paper. For your attention, I present the reviewer’s comments with bullet points and provide my responses in the order in which they appeared in the reviews.

The Authors decided to explore the association between BMI and work time. It is an interesting approach to the socioeconomic burden of overweight and obesity, which is not often taken in literature. The study population included a relatively small, but acceptable number of individuals and was well balanced between the sexes.

Detailed comments regarding particular parts of the text are provided below:

Abstract:

  • line 17 – “and point up the need to” – maybe it should be ‘’point out”;

-It was corrected as the referee suggested.

Introduction:

  • 64-65 – there is a statement suggesting that the work hours were used as an “additional productivity measure”; however, the title of the article suggests that this was the main measure analysed in the study; thus, in the Reviewer’s opinion, writing it in a more precise way would be helpful;

Line 69- “additional productivity measure” was replaced with “primary productivity measure” since work hour is the primary outcome as the referee pointed out. I thank the referee for raising this important issue.

Material and methods:

  • in the Reviewer’s opinion the term “high skilled jobs” should be defined more precisely; maybe some examples of jobs included in this category will help the reader to fully understand the categorisation into this group;

-Examples of the high skill job were provided in Section 2.1 Data Collection.

Line 131- “Occupation categories were divided into low-skill and high-skill jobs based on the BCS social class grouping. A high-skill occupation was defined as a professional, technical, or managerial job. Some of the jobs listed in this occupation category are chief executive, doctor, lawyer, nurse, actor, journalist, engineer, and teacher.”

  • data on body height and weight of the participants were reported by the participants themselves, which is a significant limitation of the study; it should be discussed in the “Discussion” part of the text; additionally, there should also be information in “Material and methods” part of the text, regarding how exactly the information on body height was collected (i.e. over the phone/ internet survey, in-person etc.)

-The BCS data were collected through face-to-face interview and self-completion. Among many variables, the information on body weight and height were collected through face-to-face interview. This method of data collection was explained in Section 2.1 Data Collection. In Section 4, the limitation of the data on body weight and height were discussed as the referee suggested.

Section 2.1:

Line 113- “The BCS data were collected through face-to-face interview and self-completion.”

Line 128- “Participants' BMIs in adulthood were derived by the BCS based on self-reported weights and heights from the interviews.”

Section 4:

Line 390- “Despite this study’s contributions described above, it has some limitations. First, the BCS data of height and weight were collected through interview and thus the data may contain measurement error although the reported heights were consistent across the surveys.”

  • another limitation of the study is the fact, that the participants were not divided based on the character of their jobs – more vs less physically demanding; there will be a significant difference between physically demanding jobs (e.g. building, carrying heavy items) and activities associated with office-type jobs in the context of the study; it should be discussed in the “Discussion” part of the text;

- As the referee pointed out, limited information on job characteristics is a critical obstacle for disclosing the mechanism behind the BMI impact on work hours. This limitation is discussed in Discussion Section.

Section 4:

Line 393- “Second, workers were divided based on their occupation category rather than job characteristics such as physically demanding job or sedentary office job. The limited information restricts further analysis for disclosing the mechanism behind the BMI impact on work hours.”

  • in the Reviewer’s opinion, the process of inclusion/ loss of participants from the study group will be clearer for the reader if it is presented in the form of a diagram rather than a full description in the text.

-Following the referee’s suggestion, a diagram was inserted in the text (line 125) as Figure 1 to illustrate the inclusion/exclusion process of study participants.

Reviewer 2 Report

Dear Authors I carefully evaluated your paper, finding it overall well written and well presented. The Authors have presented the results of an evaluation of the impact of BMI on working hours. The results showed that BMI has significant impacts on work hours but the pattern 9 is different by gender and BMI groups. The authors have deepened a current and crucial theme that needs to be properly investigated. The theme is topical, but there are some minor concerns. The language and the grammar are good; perform just a careful re-reading in order to fix minimal inaccuracies. The overall structure of the manuscript is also accurate, and the fluency of the reading is overall good along the whole text. Abstract and keywords are discreet both in terms of appropriateness of context and the purpose of study. The aim of your research has been properly highlighted. Introduction is of good quality. Study motivation is relevant, and a clear literature gap is properly reported. The theoretical background is strong, and the need to investigate this issue is clearly documented. Statistical analysis is appropriate and relevant. Results are clear and well presented. Discussion. It needs some improvements. Authors should try to explain the reason why there is a marked difference in the impact of BMI on working hours by gender. Is there a plausible or putative biological mechanism at the basis of this finding? Moreover, practical implications (e.g. proper management of gender related health problems in the workplace) should be reported. Finally, discuss the generalizability of your finding. Overall, the results are poorly discussed. Best Regards

Author Response

Point-by-point responses to Reviewer 2

I thank the referee for reviewing my paper "Heterogeneous Impacts of Body Mass Index on Work Hours." I appreciate the helpful feedback from the reviewer. I carefully addressed the reviewer’s comments and suggestions and incorporated them in the revised version of my paper. For your attention, I present the reviewer’s comments with bullet points and provide my responses and the revised parts highlighted with blue letters in the order in which they appeared in the reviews.

Dear Authors I carefully evaluated your paper, finding it overall well written and well presented. The Authors have presented the results of an evaluation of the impact of BMI on working hours. The results showed that BMI has significant impacts on work hours but the pattern 9 is different by gender and BMI groups. The authors have deepened a current and crucial theme that needs to be properly investigated. The theme is topical, but there are some minor concerns. The language and the grammar are good; perform just a careful re-reading in order to fix minimal inaccuracies. The overall structure of the manuscript is also accurate, and the fluency of the reading is overall good along the whole text. Abstract and keywords are discreet both in terms of appropriateness of context and the purpose of study. The aim of your research has been properly highlighted. Introduction is of good quality. Study motivation is relevant, and a clear literature gap is properly reported. The theoretical background is strong, and the need to investigate this issue is clearly documented. Statistical analysis is appropriate and relevant. Results are clear and well presented.

Discussion. It needs some improvements.

  • Authors should try to explain the reason why there is a marked difference in the impact of BMI on working hours by gender. Is there a plausible or putative biological mechanism at the basis of this finding?

-In order to explain the difference in the BMI impact on work hours, two possible explanations were provided. First, there is a clear difference in work hours between men and women. Men tend to work full-time with at least 35 hours per week whereas about half of women work less than 35 hours. There may be gender-specific social norms in work hours as presented in the BCS data. In the presence of restricted work hour pattern, it is plausible that men are less likely to respond to changes in BMI than women. The other possible explanation is biological reason as the referee pointed out. For example, in the UK bariatric surgery is more common among women than men as reported in national survey of hospital episodes (Gatineau et al., 2014). These possible reasons were explained in Discussion Section.

Line 345- “The notable finding is that the nonlinear impact of BMI is evident in women but not in men. Two possible explanations are proposed here. One reason for the insignificant effect of BMI on men’s work hours is the relatively small variation in work hours among men. Most men worked at least 35 hours regardless of skill level and only a few (2.68%) worked less than 35 hours. Half of women (50.36%), on the other hand, worked less than 35 hours. Therefore, in the presence of restricted work hour pattern for male workers, it is plausible that men are less likely to respond to an increase of BMI and a consequent change in health status compared to women. The other possible explanation for different BMI impact by gender is biological differences between men and women. Women may respond differently than men to changes in health conditions associated with obesity. For example, the majority of hospital episodes for bariatric surgery in England of the UK has been taken up by women between 2000 and 2012 (e.g. 82.4% in 2000/2001 and 76.3% in 2011/2012) according to Gatineau et al.[6].”   

  • Moreover, practical implications (e.g. proper management of gender related health problems in the workplace) should be reported.

-Following the referee’s suggestion, more discussion on practical implications was provided in Discussion Section.

Line 378- “Third, the gender and skill level-specific threshold of BMI broadly corresponded with the clinical obesity cutoff suggested by the World Health Organization [31], substantiating BMI's important role in classifying excessive body weight across different groups. In this sense, the current study provides practical guidelines based on BMI thresholds and job skill level to identify workers who are at greater risk of bearing economic burden from obesity. Implementing public health policies that target the most vulnerable group of workers through proper management of gender-related health problems will effectively minimize obesity costs in the workplace.”

  • Finally, discuss the generalizability of your finding. Overall, the results are poorly discussed.

-For generalizability of the findings from this study, more discussion on how the results from this study are in line with findings from previous literature was provided in several parts of the Discussion Section.

Line 359- “The results of nonlinear relationship between BMI and work hours and differences in the BMI impact across gender are in line with previous findings on other productivity measures. For example, studies by Gregory and Ruhm [19], Kan and Lee [20], and Pinkston [26] examined workers in the US and showed negative effects of BMI on hourly wages for women in particular with evidence of nonlinearity. Kline and Tobias [18] also found negative nonlinear impact of BMI on wages of men and women in the UK. Several studies around the globe also found significant relationships between BMI in the obesity range and work-related limitations including absenteeism and presenteeism. DiBonaventura et al. [27], Finkelstein et al. [11], and Gates et al. [14] examined workers in the US, Bustillos et al. [13] for Canada, and Kudel et al. [28] for Brazil to derive negative impact of BMI or obesity on workplace productivity.”

Line 371- “This study’s findings have several implications. First, given the previous evidence of negative BMI impact on hourly wages, the adverse effects of higher BMI on work hours suggest that a higher BMI can doubly increase the economic burden, as hourly wages and work hours determine labor income. A recent study from Denmark corroborates this notion. Kjellberg et al. [29] found that a higher BMI above 30 was negatively associated with mean annual income.”

Line 386- “Finally, this study focused on work hours based on British data, but the threshold approach to detect nonlinear relationship can be applied to various health or economic outcomes in other part of the world including Asia, where obesity is an emerging health issue.”
